# The Peptide Venom Composition of the Fierce Stinging Ant *Tetraponera aethiops* (Formicidae: Pseudomyrmecinae)

**DOI:** 10.3390/toxins11120732

**Published:** 2019-12-14

**Authors:** Valentine Barassé, Axel Touchard, Nathan Téné, Maurice Tindo, Martin Kenne, Christophe Klopp, Alain Dejean, Elsa Bonnafé, Michel Treilhou

**Affiliations:** 1EA-7417, Institut National Universitaire Champollion, Place de Verdun, 81012 Albi, France; axel.touchard2@gmail.com (A.T.); nathan.tene@univ-jfc.fr (N.T.); elsa.bonnafe@univ-jfc.fr (E.B.); michel.treilhou@univ-jfc.fr (M.T.); 2Laboratory of Animal Biology and Physiology, Faculty of Science, University of Douala, P.O. Box 24157, Douala, Cameroon; tindodouala@yahoo.com (M.T.); medoum68@yahoo.fr (M.K.); 3Unité de Mathématique et Informatique Appliquées de Toulouse, Genotoul Bioinfo, UR0875, INRA Toulouse, 31326 Castanet-Tolosan, France; christophe.klopp@inra.fr; 4Ecolab, Université de Toulouse, CNRS, INPT, UPS, 31400 Toulouse, France; alain.dejean@wanadoo.fr; 5CNRS, UMR EcoFoG, AgroParisTech, Cirad, INRA, Université des Antilles, Université de Guyane, 97310 Kourou, France

**Keywords:** defensive venom, dimeric peptides, peptidome, *Tetraponera aethiops*

## Abstract

In the mutualisms involving certain pseudomyrmicine ants and different myrmecophytes (i.e., plants sheltering colonies of specialized “plant-ant” species in hollow structures), the ant venom contributes to the host plant biotic defenses by inducing the rapid paralysis of defoliating insects and causing intense pain to browsing mammals. Using integrated transcriptomic and proteomic approaches, we identified the venom peptidome of the plant-ant *Tetraponera aethiops* (Pseudomyrmecinae). The transcriptomic analysis of its venom glands revealed that 40% of the expressed contigs encoded only seven peptide precursors related to the ant venom peptides from the A-superfamily. Among the 12 peptide masses detected by liquid chromatography-mass spectrometry (LC–MS), nine mature peptide sequences were characterized and confirmed through proteomic analysis. These venom peptides, called pseudomyrmecitoxins (PSDTX), share amino acid sequence identities with myrmeciitoxins known for their dual offensive and defensive functions on both insects and mammals. Furthermore, we demonstrated through reduction/alkylation of the crude venom that four PSDTXs were homo- and heterodimeric. Thus, we provide the first insights into the defensive venom composition of the ant genus *Tetraponera* indicative of a streamlined peptidome.

## 1. Introduction

Venoms are biochemical arsenals developed by animals to defend themselves and/or capture prey [1]. Studies on arthropod (e.g., scorpions, spiders, centipedes, and insects) venoms show a majority of proteins and peptides which exhibit variable amino acid sequences and tridimensional structures in their mature form [2,3]. Although ants are a dominant feature in terms of number of individuals and biomass in most terrestrial ecosystems, few extensive studies have been conducted on their venoms. Most of these studies have focused on emblematic and/or problematic species whose stings are painful and can cause allergies [4,5,6], with the major goal being to relieve the symptoms prompted by these venoms. Consequently, the allergenic venoms of fire ants of the genus *Solenopsis* (Myrmicinae), *Pachycondyla* spp. (Ponerinae), or Australian *Myrmecia* (Myrmeciinae) have especially been studied [5,6,7,8,9]. Also, the painful venoms of the bullet ant *Paraponera clavata* and of ponerine ants of the genera *Dinoponera* and *Neoponera* have been the subject of several studies which led to the isolation and characterization of tens of bioactive peptides [10,11].

In the past, the investigation of ant venoms was greatly limited by the difficulty in gathering large amounts of venom due to the small size of these insects. However, the use of new techniques to collect venoms now permits large amounts of ant venom to be quickly obtained [12,13]. Furthermore, the use of multi-omics strategies to study ant venoms recently revealed the whole peptidome of three ant species belonging to the subfamilies Ponerinae, Myrmicinae, and Myrmeciinae, allowing a high-throughput identification of novel peptides. The rise of such an integrated methodology now allows us to undertake the in-depth exploration of new ant venoms. The comprehensive inclusion of ant species from different subfamilies and with different ecologies should enhance our understanding of the molecular diversification of ant venom peptidomes and then lead to further discoveries.

To date, only six peptides have previously been identified from Pseudomyrmecinae venoms [14], but earlier studies revealed that venoms from this subfamily are rich in peptides even though the peptide composition is greatly influenced by both the hunting habits and the nesting mode of the species [15]. Thus, in order to contribute to the cataloguing of peptide toxins from the main lineage of ants, the present study focuses on the description of the venom peptidome of *Tetraponera aethiops*, Smith, F., 1877 (Pseudomyrmecinae). *Tetraponera aethiops* is an arboreal plant-ant involved in an obligatory mutualistic association with the myrmecophyte *Barteria fistulosa* (Passifloraceae), myrmecophytes being plants sheltering colonies of specialized “plant-ant” species in hollow structures called domatia [16]. Here, the colonies, housed in the plant’s hollow twigs, feed on honeydew exudates from coccids tended inside the domatia and on symbiotic fungi. In return, the workers fiercely protect the host myrmecophyte from competing vines, and herbivorous insects and mammals [17,18,19]. Known to be very painful to humans, *T. aethiops* venom is rarely used to capture prey. Instead, it is essentially employed to defend its host tree since most of the stung, paralyzed, or killed defoliating insects are discarded rather than being retrieved to be eaten [20]. We first aimed to verify if the venom of this species contains dimeric peptides as is known for the pseudomyrmecine plant-ant *Pseudomyrmex* [14,15]. Based on previous investigations on defensive venoms [1], we hypothesized that such a specialized mutualistic interaction between *T. aethiops* and *B. fistulosa* is likely to have strongly affected the ant venom composition in favor of the production of dimeric peptides to ensure host plant protection rather than prey capture.

## 2. Results

### 2.1. Mass Spectrometry of Tetraponera Aethiops Venom

The initial LC–MS analysis was performed on an LCQ-ion trap Advantage mass spectrometer in order to establish the list of the masses of the most abundant peptides in *Tetraponera aethiops* venom (Table 1). Before sequence determination, each peptide detected was tentatively named based on the initials of the genus and species followed by their molecular weight (i.e., Ta-XXXX) as described by Johnson et al. [21]. The fully sequenced peptides were then renamed in accordance with the nomenclature developed for venom peptides [22] and we used pseudomyrmecitoxin (PSDTX) to define the venom peptides of pseudomyrmecine ants [23]. The total ion chromatogram (TIC) analysis of the venom of *T. aethiops* revealed several peaks with 12 masses corresponding to peptides (Figure 1). These peptides were relatively large, exhibited molecular weights ranging from 2662 to 5774 Da, and eluted at retention times between 14.55 and 50.07 min—which is equivalent to 15 and 56% acetonitrile in the mobile phase. The relative abundance of these peptides is reported in Table 1. The venom peptidome was mostly dominated by a single peptide, U_4_-PSDTX-Ta1a, which accounts for 66.41% of the venom peptide composition. Interestingly, peaks eluting from 31 to 45 min may depend on non-peptide compounds such as large proteins (MW > 10 KDa). The peak eluting from 54 to 55 min was dominated by several low molecular weight compounds co-eluting with the twelfth peptide (U_5_-PSDTX-Ta1a) found at trace levels (Table 1).

The crude venom was submitted to LC–MS/MS using an Orbitrap mass spectrometer for a de novo sequencing which yielded 266 sequence tags with an ALC score higher than 60%. These MS/MS sequences were then used to confirm and identify the putative peptide sequences inferred from the transcriptomic analysis of the *T. aethiops* venom gland transcriptome. Orbitrap mass spectrometry analysis confirmed the presence of U_5_-PSDTX-Ta1a in this venom, albeit in a low amount.

The mass spectrometry proteomics data have been deposited to the ProteomeXchange Consortium [24] via the PRIDE [25] partner repository with the dataset identifier PXD016337.

### 2.2. Venom Gland Transcriptome and Predictive Precursor Sequences

The RNA sequencing of the *T. aethiops* venom apparatus resulted in the acquisition of 188,907,225 demultiplexed raw reads with a length of 150-bp. They were assembled de novo using Oases graph assembler, which resulted in 16,047 contigs. Among them, 230 transcripts were expressed to a frequency higher than or equal to 100,000 hits and were subsequently addressed using the NCBI blastp program for functional annotations (Appendix A). These annotated transcripts were classified into four categories (i.e., venom peptides, venom proteins, protein maturation, and others; Figure 2). This permitted us to deduce that venom peptide toxins have high transcription levels accounting for 40% of the most expressed transcripts by the venom glands (400,732 Reads Per Million (RPM)). Thus, we identified six putative venom peptide precursors. They shared an average of 52% identity, allowing us to define the following consensus sequence from the alignment of the prepro-regions: MXLSYXXLXLXVXFXLAIXFXPXXXAXAXSVGMADAEXXALAESXANALADAXP.

*Tetraponera aethiops* peptide precursors seemed to be related to other venom peptide precursors reported for ants. Indeed, this consensus sequence shared 43%, 35%, 52%, and 48% sequence identity with consensus prepro-sequences obtained from studies on *Odontomachus monticola* [26], *Myrmecia pilosula* [27], *Myrmecia gulosa* [28], and *Tetramorium bicarinatum* venoms [29], respectively (Figure 3). Unlike prepro-regions, mature regions of these precursors exhibited much more intra- and interspecific amino acid sequence variabilities (Figure 4). The complete cDNA sequences of venom peptide precursors from *Tetraponera aethiops* venom were submitted to GenBank (for accession numbers, see the legend to Figure 4).

Several venom proteins were also expressed in *T. aethiops* venom glands including phospholipases and venom allergens accounting for 17% of the venom gland expression (164,978 RPMs). Additionally, 3% of the transcripts relied on protein maturation (33,782 RPMs) with enzymes implicated in the formation of disulfide bonds such as protein disulfide isomerase (3347 RPMs). In keeping with protein maturation, an angiotensin-converting enzyme (521 RPMs) and two types of venom Dipeptidylpeptidase IV (2356 RPMs and 736 RPMs) were identified within the transcripts. These enzymes are likely involved in the processing of bioactive peptides [30,31].

RNAseq data are available on the European Nucleotide Archive website (https://www.ebi.ac.uk/ena) under the following study accession number: PRJEB35699.

### 2.3. Molecular Features of Pseudomyrmecitoxins

The combination of LC–MS analysis, de novo sequencing-based mass spectrometry and the RNA sequencing of the venom glands allowed us to assign five linear peptide sequences (i.e., U_1_-PSDTX-Ta1a, U_2_-PSDTX-Ta1a, U_3_-PSDTX-Ta1a, U_4_-PSDTX-Ta1a and U_5_-PSDTX-Ta1) to the twelve masses initially detected through LC–MS (Table 2). Additionally, the transcriptomic data led to the identification of three highly expressed peptide transcripts (i.e., 178,027; 148,472, and 144,674 RPMs) having very similar sequences with calculated masses in the range 2723–2877 Da and bearing an odd number of cysteines (i.e., one and three).

Since several dimeric peptides have been found in ant venoms [32], particularly for the Pseudomyrmecinae [14,15,33], we hypothesized that these three transcripts encoded dimeric peptide subunits. Based on this hypothesis, we calculated the theoretical masses of these presumed homo/heterodimeric peptides. Indeed, the calculated masses (i.e., 5438.60; 5750.94; 5608.78; 5680.86 Da) were detected in the total ion chromatogram of the crude venom corresponding to the pseudomyrmecitoxins Ta-5438, Ta-5750, Ta-5608, Ta-5680, and eluting from 23.64 to 26.75 min (Table 1). In order to confirm the presence of these four dimeric peptides, the crude venom was reduced using dithiothreitol (DTT) and then alkylated with iodoacetamide (IA). Both reduced and reduced/alkylated venoms were submitted to LC–MS analysis on an LCQ-ion trap Advantage mass spectrometer following the same elution conditions as crude venom. The comparisons of both chromatograms and spectra before and after reduction/alkylation support our hypothesis of dimeric features. As these dimeric pseudomyrmecitoxins shared a high percentage identity (an average of 57%), we used the same subscript to denote the ‘unknown’ activity descriptor prefix (U_2_). The homodimeric U_2_-PSDTX-Ta1b toxin (Ta-5438) is structured by three disulfide bonds, but further investigations are required to know whether cysteines form three interchain disulfide bonds or one interchain plus two intrachain disulfide bonds. Both homodimer U_2_-PSDTX-Ta1a (Ta-5750) and homodimer U_2_-PSDTX-Ta1c (Ta-5608) are linked together by a single disulfide bond. The heterodimeric peptide Ta-5680 was formed of two very similar chains (i.e., U_2_-PSDTX-Ta1a and U_2_-PSDTX-Ta1c) linked together by one disulfide bond (Figure 5). It should also be noted that the monomer form of U_2_-PSDTX-Ta1a was detected in the venom in a relatively small proportion (0.28% of the whole venom peptidome).

Overall, except for U_5_-PSDTX-Ta1a, all the pseudomyrmecitoxins in the *T. aethiops* venom are polycationic and basic (pI ranging from 8.26 to 10.84) having a net charge ranging from 1 to 5.8. Both peptide U_4_- and U_5_-PSDTX-Ta1a possess a high proportion of hydrophobic residues (Table 2). Furthermore, several pseudomyrmecitoxins shared a high percentage identity with previously reported ant venom peptides. Thus, chains from both homo- and heterodimers (i.e., U_2_-PSDTX-Ta1a-c) shared an average of 30% identity with the A-chain of M-MIITX-Mp2a from *Myrmecia pilosula*. In addition, two other pseudomyrmecitoxins (i.e., U_3_-PSDTX-Ta1a and U_4_-PSDTX-Ta1a) shared sequence identities with myrmeciitoxins previously found for *Myrmecia gulosa* [28], poneratoxins found for *Neoponera goeldii* [34], and myrmicitoxins found for *Tetramorium bicarinatum* [29] (Figure 6). The structural prediction performed on the PepFold3 server suggested that all pseudomyrmecitoxins adopt secondary structures dominated by α-helices [35].

## 3. Discussion

Defensive venoms such as those of bees or fishes are arguably comprised of conserved toxins acting primarily to trigger pain and being less complex in composition than predatory venoms [1]. The venomic investigation conducted in this study revealed that the defensive venom peptidome of *T. aethiops* is streamlined, containing only twelve peptides, with one being very dominant accounting for 66.41% of the overall peptidome (Figure 1, Table 1). This venom peptide diversity is obviously lower than the highly complex predatory venoms of spiders [37], scorpions [38], centipedes [39], or cone snails [40]. The venom peptidome of *T. aethiops* is also substantially less complex than the predatory ant *Tetramorium bicarinatum* venom peptidome, which is encoded by 37 peptide genes belonging to three superfamilies [29]. However, this peptide diversity is similar to those of other ants with a dual use of offensive/defensive venom such as *Myrmecia gulosa* [28] and *Odontomachus monticola* [26]. Different venom compositions have also been noted in the subfamily Pseudomyrmecinae according to a mass spectrometry-based investigation. Indeed, the predatory ground-dwelling species *Pseudomyrmex termitarius* has a venom composed of 87 linear peptides, whereas the venoms of the arboreal *P. gracilis* and the plant-ant *P. penetrator* are composed of only 23 and 26 peptides, respectively, certain of them with disulfide bonds and dimeric features [15].

Here we also demonstrate the presence of a set of homo- and heterodimeric peptides in the *T. aethiops* venom which is consistent with previous examinations of pseudomyrmicine venoms from the Paleotropical genus *Tetraponera* [33] and the Neotropical genus *Pseudomyrmex* genera [14,15]. Interestingly, none of the dimeric pseudomyrmecitoxins characterized in *T. aethiops* venom show sequence identity with known heterodimeric pseudomyrmecitoxins found in *Pseudomyrmex triplarinus* venom (see [14]). Nevertheless, our data revealed sequence identity with a dimeric myrmeciitoxin from *Myrmecia pilosula*, M-MIITX-Mp2a: 36%, 29%, and 25% for U_2_-PSDTX-Ta1a, U_2_-PSDTX-Ta1b, and U_2_-PSDTX-Ta1c, respectively (Figure 6; See [27,41]). Altogether, these multiple dimeric peptides found in the genera *Myrmecia*, *Pseudomyrmex* and *Tetraponera*, suggest that dimeric scaffolds are recurrent in the phylogenetic clade Pseudomyrmecinae/Myrmeciinae, with these two subfamilies being close phylogenically [42]. Over a broader scale, dimeric venom peptides were found in the venom of several ant subfamilies (i.e., Pseudomyrmecinae, Ectatomminae, Myrmeciinae, and Ponerinae) [23,27,28,32,33,43] while this structural feature was only occasionally noted in the peptidome of other venomous animals [44,45,46]. Interestingly, these ant species possessing dimeric peptide toxins in their venom are also well-known for the intense pain induced by their sting. This observation raises the matter of the evolutionary advantage of such dimeric scaffolds for ant venoms that might be related to the defense of the colony and extended to the host myrmecophyte for plant-ants ([15], this study).

In terms of primary sequence, several of the pseudomyrmecitoxins described here have sequence identities with other ant venom dimeric peptides which are known for their pain-inducing properties. Actually, each chain of dimeric U_2_-PSDTXs has a sequence identity with the A-chain of the M-MIITX-Mp2a, the heterodimeric pilosulin discovered in *Myrmecia pilosula* venom (Figure 6). Indeed, Dekan et al. (2017) showed that M-MIITX-Mp2a, a membrane-disrupting peptide displaying broad-spectrum antimicrobial and nociceptive properties, also induces a concentration-dependent transient increase in the intracellular Ca^2+^ in neuronal cells eliciting spontaneous pain in mice [36]. Along the same line, U_4_-PSDTX-Ta1a, the major linear peptide in *T. aethiops* venom has 45% sequence identity with the major linear peptide of *Myrmecia gulosa* venom, MIITX_1_-Mg1a (Figure 6). Robinson et al. (2018) showed that this peptide possesses a membrane disrupting activity capable of causing a leak in membrane ion conductance, thus altering membrane potential and triggering neuronal depolarization. On the one hand, it activates mammalian sensory neurons, which is consistent with the capacity to produce pain. On the other hand, MIITX_1_-Mg1a is also able to incapacitate arthropods, demonstrating its multifunctional role [28]. Presumably, these sequence identities with pain-inducing peptides might explain the extremely painful *T. aethiops* stings even though the functional characterization of pseudomyrmecitoxins is obviously required to remove any doubt.

The defensive function of the venom is likely one of the major factors that favored the rise of sociality in hymenopterans, driving the evolution of the venom toward the protection of the colony from predators [47]. The present investigation provides another piece to the evolutive puzzle of Hymenoptera venoms where polycationic, amphiphilic, α-helical peptides, which are sometimes dimeric, seems to play a major role for colony protection.

## 4. Conclusions

In this study, we investigated the venom peptidome of *Tetraponera aethiops* which is mainly used for defensive purposes. Our data revealed that the *T. aethiops* venom peptidome contains twelve venom peptides, among which nine were identified through the combination of transcriptomic and proteomic data. We hypothesized that the imposed selective pressures to deter predators have led toward the simplification of *T. aethiops* venom likely dominated by pain-inducing toxins. *Tetraponera aethiops* venom peptides, which are either linear or dimeric, possess substantial sequence identities with myrmeciitoxins previously described as both pain-inducing and insecticidal. Nevertheless, the functional characterization of *T. aethiops* venom pseudomyrmecitoxins is needed to come to a conclusion on their biological significance.

## 5. Materials and Methods

### 5.1. Collection of the Ants and Preparation of the Venom Samples

*Tetraponera aethiops* workers were collected in Cameroon, near Douala (N3°53.310′ E10°44.918′) in 2016 and 2018. The ants’ venom sacs were dissected and placed together in water containing 1% formic acid (*v*/*v*) and the membranes disrupted using ultrasonic waves for 2 min. Then, the samples were centrifuged for 5 min at 14,400 rpm, the supernatant was collected and dried with a speed vacuum prior to storage at −20 °C until use. Four venom samples containing 14, 19, 20, and 29 venom sacs were used for all the proteomic analyses.

### 5.2. Mass Spectrometry Analysis

A preliminary LC–MS analysis of the crude venom was carried out on the LCQ-Ion trap Surveyor equipped with an ESI-LC system Advantage (ThermoFisher Scientific, Courtabœuf, France). Peptides were separated using an Acclaim RSLC C_18_ column (2.2 µm; 2.1 × 150 mm; Thermofisher, France). The mobile phase was a gradient prepared from 0.1% aqueous formic acid (solvent A) and 0.1% formic acid in acetonitrile (solvent B). The peptides were eluted using a linear gradient from 0 to 50% of solvent B during 45 min, then from 50 to 100% during 10 min, and finally held for 5 min at a 250 µL min^−1^ flow rate. The electrospray ionization mass spectrometry detection was performed in positive mode with the following optimized parameters: the capillary temperature was set at 300 °C, the spray voltage was 4.5 kV, and the sheath gas and auxiliary gas were set at 50 and 10 psi, respectively. The acquisition range was from 100 to 2000 *m*/*z*. The area value of each peak corresponding to a peptide was manually integrated using the peak ion extraction function in Xcalibur software (version 4.0, ThermoFisher Scientific, Courtabœuf, France). The relative peak area indicates the contribution of each peptide to all the peptides identified in the venom, providing a measure of relative abundance.

### 5.3. Disulfide Bond Reduction and Alkylation

The presence of dimeric peptides in *T. aethiops* venom was determined via chemical reduction/alkylation of crude venom and subsequent LC–MS analysis. Disulfide reduction was achieved by mixing 40 µL of crude venom (six venom reservoirs) incubated with 40 µL of 100 mM ammonium bicarbonate buffer (pH 8) containing 10 mM dithiothreitol (DTT) for 30 min at 56 °C. Then, 40 µL of the reduced venom was analyzed through LC–MS to identify the dimeric peptides. Finally, the remaining 40 µL of the reduced venom was alkylated by adding 1.5 µL of 0.5 M iodoacetamide (IA) for 90 min at room temperature in the dark. As chemical reduction/alkylation results in a mass increase of 57 Da for each cysteine, the examination of mass shifts in the mass spectra of both reduced and reduced/alkylated samples permitted us to determine the number of disulfide bonds in the corresponding peptides.

### 5.4. De novo Orbitrap Mass Spectrometry-Based Sequencing

Crude venom was re-suspended in water and then desalted using ZipTip^®^ C_18_ (Merck Millipore, Burlington, VT, USA) after adding trifluoroacetic acid at a final concentration of 0.5%. Then, the venom sample was subjected to de novo sequencing using a Q-Exactive Plus mass spectrometer coupled to a Nano-LC Proxeon 1000 (ThermoFisher Scientific, Waltham, MA, USA). Peptides were separated through chromatography with the following parameters: Acclaim PepMap100 C_18_ pre-column (2 cm, 75 μm i.d., 3 μm, 100 Å), Pepmap-RSLC Proxeon C_18_ column (50 cm, 75 μm i.d., 2 μm, 100 Å), 300 nL min^−1^ flow rate, a 98 min gradient from 95% solvent A (water, 0.1% formic acid) to 35% solvent B (99.9% acetonitrile, 0.1% formic acid) for a total time of 2 h. Peptides were analyzed in the Orbitrap cell, at a resolution of 120,000, with a mass range of *m*/*z* 350–1550. Fragments were obtained through high collision-induced dissociation (HCD) activation with a collisional energy of 27%. Data were acquired in the Orbitrap cell in a Top20 mode, at a resolution of 17,500. For the identification step, all MS and MS/MS data were processed with an in-house Peaks software (BSI, version 6.0) to perform de novo sequencing. The mass tolerance was set to 10 ppm for precursor ions and 0.02 Da for fragments. The following modifications were allowed: oxidation (Met) and pyroglutamic acid (Glu). De novo peptide sequences with Average Local Confidence (ALC) higher than 60% were used for the peptide identifications.

### 5.5. Direct Sequencing of Venom Gland RNA

Venom glands and sacs from 20 live ant workers, anesthetized by cooling, were dissected in a PBS solution. Each tissue was immediately placed in 500 µL of TRIzol reagent (Invitrogen, Carlsbad, CA, USA) and total RNAs were extracted afterward using the RNeasy Micro Kit (Qiagen) according to manufacturer’s instructions. Contaminating genomic DNA was removed using a DNA-free kit (Applied Biosystem) according to the manufacturer’s instructions. RNA quantity was evaluated using a nanodrop and a bioanalyzer (Nanodrop 2000, ThermoFisher Scientific; Agilent 2100 Bioanalyzer System). RNAseq was performed at the GeT-PlaGe core facility, INRA Toulouse, France. RNA-seq libraries were prepared according to Illumina’s protocols using the Illumina TruSeq Stranded mRNA sample prep kit to analyze mRNA. Briefly, mRNA was selected using poly-T beads. Then, the RNA was fragmented to generate double stranded cDNA and adaptors were ligated to be sequenced. Eleven cycles of PCR were applied to amplify the libraries. Library quality was assessed using a Fragment Analyser, and the libraries were quantified through qPCR using the Kapa Library Quantification Kit. RNA-seq experiments were performed on an Illumina HiSeq3000 using a paired-end read length of 2 × 150 pb with the Illumina HiSeq3000 sequencing kits.

### 5.6. Bioinformatic Tools

#### 5.6.1. Contigs Quantification

The read pairs were assembled twice with drap (version 1.9.1) [48] using the de Bruijn graph assembler called oases (parameter: –dbg oases). The assembly metrics were produced with the assemblathon_stats.pl scripts. Raw reads were aligned on the contigs with bwa mem (version 0.7.12-r1039) [49] using the default parameters and the alignment files were sorted, compressed, and indexed with samtools view, sort, and index (version: 1.3.1) using the default parameters [50]. The quantification files were generated with samtools idxstats (version: 1.3.1), giving us the length of each contig in base pairs along with the number of hits, corresponding to the number of sequences from RNAseq reads which aligned on a given contig.

To calculate the expression rate of transcripts discovered in the venom gland transcriptome, the appellation Reads Per Million (RPM) was chosen over the traditionally used Transcripts count Per Million (TPM) value. Indeed, TPM calculation takes into account the length of the transcript, or in absence of reference genome, of the contig. However, for contigs containing the venom peptides open reading frames (ORF), the assembler often generates overextended contigs (see Appendix A). Thus, the expression rate of the short venom peptides transcripts would be underestimated with TPM [51]. So we calculated the RPM value for each transcript of interest in two steps: (i) by dividing the number of aligned reads for each contig by the total number of million reads aligned for the sample, and (ii) by summing up the obtained values for each contig encoding the transcript when several contigs represent the same peptide.

#### 5.6.2. Precursor Identification and Mature Sequences

RNAseq data were translated using a translate program command lines (emboss package, command line: transeq) in order to obtain the potential Open Reading Frames. Then, the fragments of sequences obtained during the de novo Orbitrap mass spectrometry-based sequencing were aligned against these data by using the command-line NCBI BLAST program (ncbi-blast-2.6.0+ package, command line: blastp, parameter: -matrix PAM30) with adapted parameters for short sequences, allowing us to find the complete peptide sequences and the name of the contigs on which they aligned.

The masses of mature peptide sequences, obtained from these different approaches, were systematically verified using the peptide mass program from ExPASy portal (https://expasy.org) and compared to those obtained through mass spectrometry. The isoelectric points, net charges and percentage of hydrophobic amino acids were calculated using ExPASy (https://web.expasy.org/compute_pi/), PepCalc (https://pepcalc.com/), and Peptide Property Calculator V3.1 (https://www.biosyn.com/peptidepropertycalculator/peptidepropertycalculator.aspx). Secondary structure predictions were conducted with the PEP-FOLD3 server [35]. Signal sequences and transmembrane domains were predicted with the phobius program available at http://phobius.sbc.su.se/. Sequence similarities were searched for using the NCBI BLAST program presented in the Uniprot server with the default parameters. Alignments were achieved and sequence identity percentages were calculated with the EMBL-EBI server and the MUSCLE program with the default parameters [52]. They were then edited using Seaview version 4.6.1 [53] and BOXSHADE version 3.2 (https://embnet.vital-it.ch/software/BOX_form.html).

#### 5.6.3. Annotation of Most Expressed Contigs

Open reading frames (≥100 amino-acids length), found by translating RNAseq data, were extracted from the most abundant contigs and then submitted to the NCBI BLAST program against the Uniprot refseq protein database on the computational cluster of the Genotoul bioinformatic facility, INRA Toulouse, France (ncbi-blast-2.6.0 + package, command line: blastp, parameter: -matrix BLOSUM62).

## Figures and Tables

**Figure 1 toxins-11-00732-f001:**
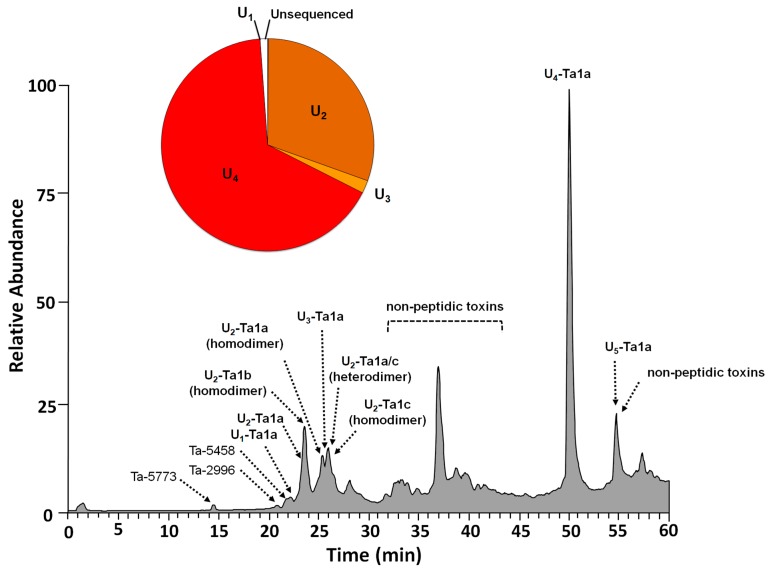
Positive mode total ion chromatogram (TIC) of *T. aethiops* venom using LCQ Advantage ESI mass spectrometer. Crude venom was separated by C_18_ RP-HPLC using an ACN gradient. The mobile phase was 0.1% aqueous formic acid (solvent A) and 0.1% formic acid in acetonitrile (solvent B). The peptides were eluted using a linear gradient from 0 to 50% of solvent B during 45 min, then from 50 to 100% during 10 min, and finally held for 5 min at a 250 µL min^−1^ flow rate. Note that ‘U_4_-PSDTX-Ta1a’ accounts for 66.41% of the venom peptide content.

**Figure 2 toxins-11-00732-f002:**
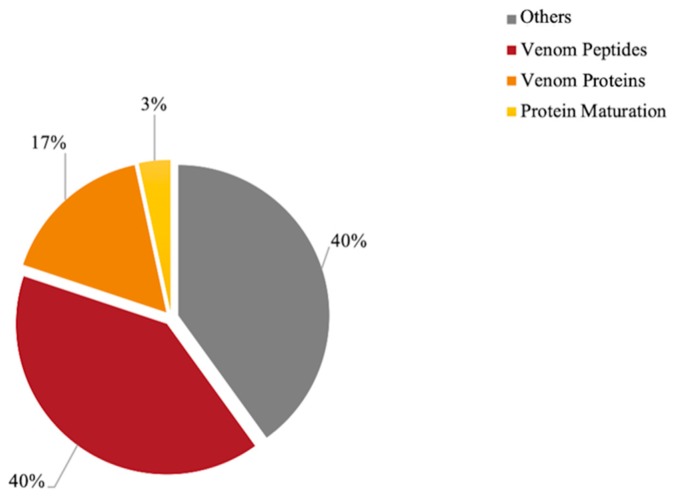
Proportions of addressed functions of the 230 most expressed transcripts (≥100,000 hits) from *T. aethiops* venom glands. Functional annotations were made with the NCBI blastp program. The category “Others” groups functions involved in cellular metabolism. Contigs coding for venom peptides accounted for 40% of the transcripts expressed by the venom glands. Seventeen percent of the transcripts coded for venom proteins such as venom allergens and phospholipases, and 3% of the transcripts were dedicated to protein maturation. The contigs names and functions are presented in detail in Appendix A.

**Figure 3 toxins-11-00732-f003:**
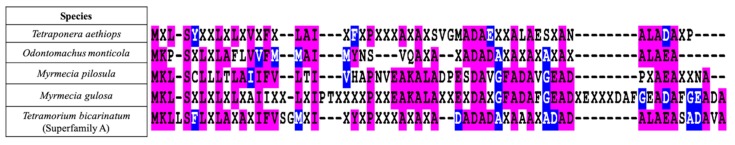
Alignment of consensus prepro sequences from pilosulin-like ant venom peptides. The alignment was generated with the Muscle program in Seaview version 4.6.1 and edited using BOXSHADE version 3.2. Identical residues are highlighted in magenta. Similar residues are highlighted in blue. The consensus sequence of the prepro-region of *Tetraponera aethiops* venom peptides shared 43%, 35%, 52%, and 48% with the consensus prepro-regions sequences obtained from previous studies on *Odontomachus monticola* [26], *Myrmecia pilosula* [27], *Myrmecia gulosa* [28], and *Tetramorium bicarinatum* (Superfamily A) [29], respectively.

**Figure 4 toxins-11-00732-f004:**
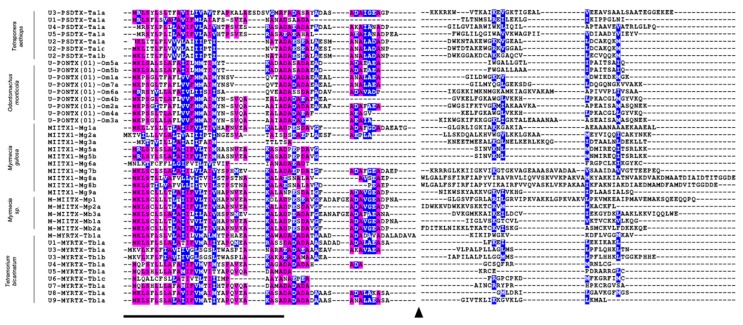
Alignments of *T. aethiops* venom peptide precursors with pilosuline-like peptides from *Odontomachus monticola* [26], *Myrmecia gulosa* [28], *Myrmecia pilosula* [27], and *Tetramorium bicarinatum* (Superfamily A) [29] venoms. Alignments were generated with the Muscle program in Seaview version 4.6.1 and edited using BOXSHADE version 3.2. The prepro- and mature regions were aligned separately. Identical residues are highlighted in magenta and similar residues are highlighted in blue. The black triangle indicates the cleavage site between the prepro-regions and the mature peptides. The black line marks the signal regions. Post-translational modifications are not shown. The prepro-regions showed themselves to be conserved, whereas mature peptide sequences were highly variable. *Tetraponera aethiops* venom peptide precursor cDNA sequences were submitted to GenBank, with the following accession numbers: U_1_-PSDTX-Ta1a (MN607166), U_2_-PSDTX-Ta1a (MN607169), U_2_-PSDTX-Ta1b (MN607170), U_2_-PSDTX-Ta1c (MN607168), U_3_-PSDTX-Ta1a (MN607165), U_4_-PSDTX-Ta1a (MN607167), and U_5_-PSDTX-Ta1a (MN607171).

**Figure 5 toxins-11-00732-f005:**
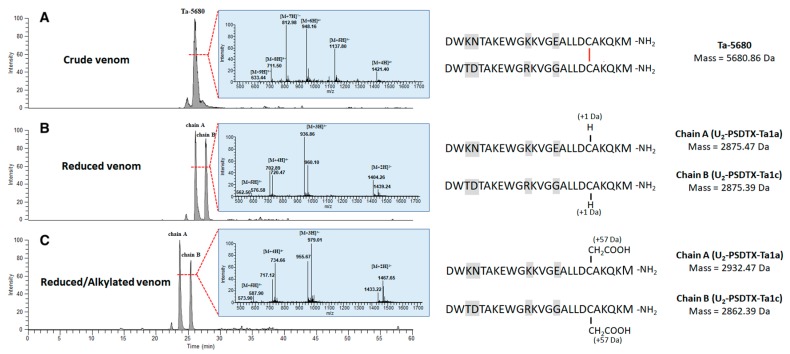
Identification of the dimeric features of the Ta-5680 peptides in *T. aethiops* venom. (**A**) Extracted-ion chromatogram and MS spectrum of the peptide Ta-5680 from the LC–MS analysis of *T. aethiops* venom before reduction/alkylation. We hypothesized that Ta-5680 is a heterodimeric peptide having the hypothetic sequence shown on the right. The distinctive residues between both monomers are highlighted in grey and the red bar represents the disulfide bond. (**B**) Comparison of chromatograms and spectra before and after reduction with DTT (Dithiothreitol) revealed the presence of two novel masses (2875.47 and 2805.39 Da) corresponding to both A and B chains while the Ta-5680 mass disappeared. (**C**) Alkylation experiment using IA (iodoacetamide) confirmed the presence of one cysteine on each alkylated monomer.

**Figure 6 toxins-11-00732-f006:**
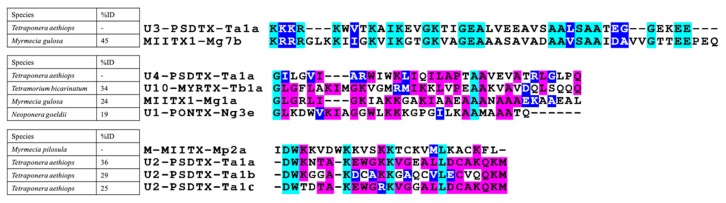
Of *Tetraponera aethiops* mature venom peptides with similar venom peptides from ant venoms [28,29,34,36]. Alignments were generated with the Muscle program in Seaview version 4.6.1 and edited using BOXSHADE version 3.2. Conserved residues are highlighted in cyan, identical residues are highlighted in magenta and similar residues are highlighted in blue.

**Table 1 toxins-11-00732-t001:** Peptide mass fingerprinting of *Tetraponera aethiops* venom. List of peptide masses detected through LC–MS using an LCQ-ion trap Advantage mass spectrometer.

Retention Time (min)	Mass (Da)	Relative Abundance (%)	Temporary Name	Peptide Toxin
14.55	5773.80	0.50	Ta-5773	
20.50	2996.28	0.03	Ta-2996	
22.18	5458.56	0.60	Ta-5458	
22.24	2663.58	0.10	Ta-2662	U_1_-PSDTX-Ta1a
23.00	2877.72	0.28	Ta-2875	U_2_-PSDTX-Ta1a
23.64	5441.80	11.98	Ta-5438	U_2_-PSDTX-Ta1b (homodimer)
25.23	5753.64	9.33	Ta-5750	U_2_-PSDTX-Ta1a (homodimer)
25.50	4470.96	1.96	Ta-4468	U_3_-PSDTX-Ta1a
26.00	5683.60	7.24	Ta-5680	U_2_-PSDTX-Ta1a/U_2_-PSDTX-Ta1c
26.75	5610.63	1.27	Ta-5608	U_2_-PSDTX-Ta1c (homodimer)
50.07	3568.28	66.41	Ta-3566	U_4_-PSDTX-Ta1a
54.93	3615.81	*	Ta-3615	U_5_-PSDTX-Ta1a

* found at trace levels.

**Table 2 toxins-11-00732-t002:** Peptide sequences in the venom of *Tetraponera aethiops*. pI, isoelectric point. For each peptide, the transcripts frequency value (Reads Per Millions; RPMs) represents the frequency sum of all assembled contigs encoding the peptide precursor. “*” denotes C-terminal amidation.

Toxin Name	Mass (Da)	RPMs	Sequence	Features	Net Charge	Hydrophobic aa (%)	pI
U_1_-PSDTX-Ta1a	2662.54	118,602	TLTNMSLREILEKLGIKIPPGLNI	Monomer	1.0	41.67	8.26
U_2_-PSDTX-Ta1a	2875.47	148,472	DWKNTAKEWGKKVGEALLDCAKQKM *	Monomer	2.9	40.00	8.99
U_2_-PSDTX-Ta1a	5750.94	148,472	DWKNTAKEWGKKVGEALLDCAKQKM *	1 S-S Homodimer	5.8	40.00	9.23
U_2_-PSDTX-Ta1b	5438.60	144,674	DWKGGAKDCAKKGAQCVLECVQQKM *	3 S-S Homodimer	5.6	44.00	8.83
U_2_-PSDTX-Ta1c	5608.78	178,027	DWTDTAKEWGRKVGGALLDCAKQKM *	1 S-S Homodimer	3.8	40.00	8.70
U_2_-PSDTX-Ta1c U_2_-PSDTX-Ta1a	5680.86	178,027 148,472	DWTDTAKEWGRKVGGALLDCAKQKM * DWKNTAKEWGKKVGEALLDCAKQKM *	Heterodimer	4.8	40.00	9.04
U_3_-PSDTX-Ta1a	4468.45	26,331	KKKRKWVTKAIKEVGKTIGEALVEEAVSAALSAATEGGEKEE	Monomer	1.0	38.10	8.27
U_4_-PSDTX-Ta1a	3566.12	126,081	GILGVIARWIWKLIQILAPTAAVEVATRLGLPQ	Monomer	2.0	60.61	10.84
U_5_-PSDTX-Ta1a	3615.97	91,424	FWGLILQGIWAVVKWAGPIIVDIAADYVIEYV *	Monomer	1.0	71.88	4.03

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
