# Peer review of "The Peptide Venom Composition of the Fierce Stinging Ant Tetraponera aethiops (Formicidae: Pseudomyrmecinae)"

_toxins, 2019, doi:10.3390/toxins11120732_

Round 1

Reviewer 1 Report

The study presents the discovery of a streamlined venom in Tetraponera aethiops based on proteomic and transcriptomic data. Though the study is well conceived and mostly well-written and can become an interesting contribution to the field of Hymenoptera venomics, it has several deficiencies. First of all, the data itself seems to be missing, no links to transcriptomic or proteomic data were provided, and as such it is impossible to verify the authors' findings. Secondly, there are several issues with the text, one of the major points is that the authors do not follow the naming convention they state they do. King et al., naming system is designed such that different isoforms and paralogs have names as part of a series. However in present paper the authors name a monomer and a homodimer that is made of that monomer as parts of the series. That is going against the rules outlined in King et al. This has to be addressed. In addition, there are several text infelicities that make it impossible to understand what the authors mean. At one point the authors misspell the name of the protein they then go on to discuss a function of, this also has to be addressed. Lastly, the method section has to be expanded in order to include all the information. At the moment it is missing some crucial parts.

Line-by-line break-down:

Line 12: “encode” should be used without prepositions, i.e. “encode only seven peptides”

Line 23/24: why do authors use semicolon instead of comma?

Line 28/29: It is unclear what the authors mean by this sentence. Especially by “structures”. If the meaning is that most of arthropods’ toxins’ groups have non-conserved consensus primary structures it has to be spelled out that way, and also – in comparison to what.

Line 32/33: “can cause allergies to overcome the symptoms” - not sure what the authors mean by that

Line 38 to 43: event though the authors are right, that in the past having high amounts of ant venom was an issue, it wasn’t resolved by omics revolution, rather it was resolved by re-thinking of the way samples are collected. For example, Fox et al., 2013 provided an easy method of collecting gram-level amounts of venom.

Line 67: “the initial” instead of “an initial”

Line 78: “accounts” or “accounted” instead of “account”. I really recommend that authors do a thorough proof-reading of the manuscript

Line 88/89: Authors are inconsistent, when they say “0 to 50% B” but then “50 to 100%”. In addition, it would be better to say “linear gradient of solvent B 0 to 50%” to avoid any confusion, because not all toxinologists are intimately familiar with HPLC slang.

Line 113: formatting issues

Line 131/132: Authors should correct the name of the enzyme as “Dipeptidylpeptidase 4”. DPP4 is a known peptide-processing enzyme in hymenopteran venom. It processes prepromelittin into melittin (Kreil, Haiml and Suchanek, 1980). This citation is a much better choice than Cito et al., 2017, which suggests that DPP4 function in venom was unknown up until 2017.

Line 144-152: The statement from the figure legend “The black triangle indicates the cleavage site between the signal regions (marked by the black line) and the mature peptides” cannot possibly be true, because the black line ends several dozens of aminoacids before the black triangle. This needs clarification.

Line 161-: Table 2, column TPM is shrunk so that it makes any comparison very difficult. The numbers need to be placed in a single line, or at least to be wrapped consistently. It is absolutely feasible to make it a single line given the size of the table and its contents, so I suggest authors do that.

Line 172: Why “agent”? Why not just “DTT”?

Ibid: Authors do not say that they have made an additional chromatogram, before jumping to discussing it. This is confusing, they should introduce the experiment before discussing it.

Line 175 and elsewhere: In the study the authors cite King et al., 2008 as a guideline for their naming system, yet the original paper states it clearly that the last letter of the name is reserved for paralogs (sic). A monomer that is a part of a dimer is not a paralog of a dimer. It is the same peptide, encoded by the same gene. Therefore it is incorrect to put two entities of different nature (monomer and dimer) as instances of the same series. If U2-PSDTX-Ta1c is made of two U2-PSDTX-Ta1a, then it should have a name that reflects that. Current naming system proposed in the study goes against the standard and is very confusing. Compare it with heterodimer neurotoxic Pla2g2 enzymes from rattlesnake venoms: subunits are called Pla2g2Ga and Pla2g2Gb, while the whole heterodimeric toxin doesn’t have a name of that sort, because it is a different entity (Dowell et al., 2016). All in all, this is a serious flaw of the study that will unnecessarily confuse future researchers that will treat U2-PSDTX-Ta1c and U2-PSDTX-Ta1a (etc) as different toxins in the databases, while in fact – from evolutionary and genetic perspective they are not. It also goes against the naming convention the authors state to be championing in the first place.

Line 222: “venom is” instead of “venomis”

Line 223-225: The authors seem to suggest that “non aggressive species” that use their venom mainly to catch prey will not have selection pressure on maintaining the defensive function of their venom. While not only there’s no data to support that claim at all, it goes against some well-known examples of “non aggressive species” having a very consistent and strong defensive venoms, e.g. tarantula hawks with the most painful stings among the known Hymenoptera. Therefore, this claim has to be substantially re-written to either clarify the statement or make it less inconsistent with what’s known on hymenopteran venoms.

235/236 and elsewhere: I suggest the authors reference their figures every time they mention something those figures are showing.

255: when basing the claim on sequence homology, it is important to state what that homology is exactly. Is it 30% homology, 50% homology or 80% homology. At the moment, the authors just state that the homology exists, and then go on to make inference based on that. Consider the following: honeybee venom acid phosphatase has 50% homology to a symphatan acid phosphatase. Yet it would be unreasonable to think that sawflies utilise it in the same manner.

269: “venomlikely” (though it is a nice word).

Method section has to be expanded. At the moment it is unclear how the authors calculated the TPM values and how did they determine the open reading frames. It is unclear what software did they use to search the peptides against the databse of toxins and whether it was done at all. Also the method of comparing proteomes with transcriptomes is not clear.

Supplementary materials are missing transcriptomic and proteomic data. It is impossible to verify the authors’ findings in absence of that.

Reviewer 2 Report

To whom it may concern:

I am happy to have reviewed you work characterizing the venom peptidome and transcriptome of Tetraponera aethiops. While I found the manuscript to be of a general high quality, I have a few suggestions for improvement.

First and foremost, the use of homology as a quantitative measurement is erroneous. It is instead a binary state; either something is homologous (i.e., stems from a common ancestor) or is not (i.e., the product of convergent or parallel evolution). This is a common mistake made by researchers not focusing on evolutionary biology, especially systematics. How were the samples for the transcriptome treated before extraction? Variations in conditions and/or timing of extractions (e.g., circadian rhythms) can have profound impacts on expression levels. Some more information concerning the pre-extraction protocols would be appreciated. BoxShade needs a citation. Versions of all software should be reported. Additionally, making your precise analytical code available through BitBucket or GitHub would be appreciated. All data and results should made available. I did not see accession numbers for anything, including sequence data. The sentence beginning on line 220 and ending on 223 needs to be rewritten for clarity.

I look forward to seeing future versions of this manuscript.

Round 2

Reviewer 1 Report

The Authors have addressed most of the criticism and have improved the manuscript sufficiently. However, the information on transcriptome sequencing data availability seems to be still missing. The Authors state that they have provided the necessary information in lines 102-103 and in Figure 4, however both of those are concerned with proteomic data only. Once the links to transcriptomic data are properly added to the text, this paper should be accepted for publication.
